# OpenReview forum: "CoaxChain: Semantically Progressive Multi-turn Jailbreak Attacks on Large Language Models"
_ICLR.cc/2026/Conference — Submitted to ICLR 2026_

### Official Review · Reviewer_7gvj · 2025-10-30

**Soundness:** 3
**Presentation:** 3
**Contribution:** 2
**Rating:** 4
**Confidence:** 5

**Summary:**

This paper introduces CoaxChain, a black-box, semantically progressive multi-turn jailbreak framework designed to systematically examine vulnerabilities in the alignment mechanisms of large language models.
The framework consists of two main components:
1. Alignment Failure Analyzer (AFA) – a white-box module operating on a locally aligned surrogate model. It performs offline gradient probing to evaluate the effectiveness of prompts, identifying those that suppress alignment-sensitive parameters without requiring risky trial-and-error interactions with the target model.
2. Semantically Progressive Prompt Generator (SPG) – a dynamic rewriting module that leverages AFA’s evaluations to select only the essential intermediate turns, constructing concise, semantically progressive dialogue sequences.
This design enhances both the effectiveness and efficiency of multi-turn jailbreaks, while providing valuable insights for developing more robust alignment defenses.

**Strengths:**

1. The paper introduces the use of critical weight sensitivity\text{Critical}(W|i) derived from a white-box surrogate model to quantify the model’s alignment sensitivity at a given conversational state. This metric is then used to guide adversarial prompt generation, effectively addressing the limitations of prior multi-turn jailbreak methods that relied on heuristic or trial-based approaches.

2. The evaluation covers not only attack success rate (ASR) but also semantic similarity (SEM), perplexity (PPL), and efficiency (measured by the number of queries). Detailed ablation studies further validate the necessity and rationality of each component, demonstrating that both AFA and the semantically progressive generation strategy are crucial for the overall performance.

3. Beyond proposing an advanced attack framework, the paper also introduces a fine-tuning–based defense mechanism, Fortify, which significantly mitigates the attack’s effectiveness. This dual-perspective contribution—offensive and defensive—enhances the paper’s overall impact and provides valuable insights for developing more robust alignment defenses.

**Weaknesses:**

1. Lack of generalization analysis of surrogate-based AFA. Although the paper claims that the Alignment Failure Analyzer (AFA) improves interpretability, the AFA is conducted entirely on that surrogate rather than the actual target mode. Because the surrogate may differ substantially in architecture and alignment mechanisms, it remains unclear why and to what extent the AFA—based on a surrogate—can reliably generalize its gradient-based alignment sensitivity assessments to unseen, closed-source targets with different architectures or alignment mechanisms.

2. Restricted and task-agnostic strategy library. The strategy pool used in the Semantically Progressive Prompt Generator (SPG) is small and predetermined for each dialogue stage, independent of task semantics. The paper lacks ablation evidence showing how the choice of strategies affects the attack success rate (ASR), or whether the same “optimal” strategy consistently performs best across different tasks and contexts. While fixing a universal optimal strategy improves reproducibility and efficiency, additional statistical justification would strengthen the claim of task-invariant optimality.

3. Writing and presentation issues. The manuscript suffers from organizational inconsistencies and repeated phrasing. Crucial components such as the deterministic renderer are left unexplained, while the discussion of predefined strategies (lines 232–233) appears misplaced within the section. These issues obscure the methodological flow and weaken the overall readability.

4. Unclear threshold stability in AFA. The paper fixes the AFA critical-weight threshold at τ = 0.4 without empirical justification or sensitivity analysis. It is uncertain how stable this threshold remains across different models or training runs, and whether small variations in τ would significantly alter which strategies are selected during adversarial prompt generation.

5. Incomplete baseline selection and insufficient discussion of related work:
The strongest baseline, ActorAttack (Oct 2024), is already outdated. Many more recent methods have surpassed it but were not included for comparison, such as [1][2][3]. In addition, the paper should provide a more comprehensive discussion of existing LLM jailbreak approaches, covering both single-turn and multi-turn attacks. Failure to include recent stronger baselines and to clearly situate the proposed method in the landscape of single- vs. multi-turn jailbreaks makes the evaluation incomplete.

[1] Yao, Yang et al. “A Mousetrap: Fooling Large Reasoning Models for Jailbreak with Chain of Iterative Chaos.” ArXiv abs/2502.15806 (2025).
 [2] Miao, Ziqi et al. “Response Attack: Exploiting Contextual Priming to Jailbreak Large Language Models.” ArXiv abs/2507.05248 (2025).
 [3] Weng, Zixuan et al. “Foot-In-The-Door: A Multi-turn Jailbreak for LLMs.” ArXiv abs/2502.19820 (2025).

**Questions:**

1. Could the authors provide either a theoretical justification or empirical analysis demonstrating the correlation between AFA outcomes on the local surrogate model and the attack success rate (ASR) observed on the target model—or on another open-source model with a different architecture and training dataset?

2. Rationality of predefined strategies. Can the authors supplement experiments to justify the use of predefined strategies for each dialogue stage, and verify whether the same stage-specific strategies remain optimal across different tasks or domains?
3. Scope of layer analysis. Why does the AFA focus exclusively on MLP layers for gradient probing? Have the authors experimented with or compared results from other architectural components (e.g., attention layers, normalization blocks)?

4. Threshold stability in AFA. How stable is the critical-weight threshold (τ = 0.4) across models and training runs? Would small variations in τ substantially change which strategies are selected during adversarial prompt generation?

5. baseline selection and discussion of related work. Can the authors include more up-to-date and comprehensive baselines? The current best-performing baseline, ActorAttack (Oct 2024), is already outdated, and several more recent methods [1][2][3] that outperform it are missing. In addition, can the authors expand their discussion of existing LLM jailbreak approaches to more thoroughly cover both single-turn and multi-turn attack methods?


[1] Yao, Yang et al. “A Mousetrap: Fooling Large Reasoning Models for Jailbreak with Chain of Iterative Chaos.” ArXiv abs/2502.15806 (2025).
 [2] Miao, Ziqi et al. “Response Attack: Exploiting Contextual Priming to Jailbreak Large Language Models.” ArXiv abs/2507.05248 (2025).
 [3] Weng, Zixuan et al. “Foot-In-The-Door: A Multi-turn Jailbreak for LLMs.” ArXiv abs/2502.19820 (2025).

---

> ### Author Response · Authors · 2025-11-21
> **Response to Reviewer 7gvj**
>
> We sincerely thank you for the detailed feedback and positive comments on Critical(W), the evaluation, and Fortify. We address your key concerns regarding AFA's generalization, the SPG strategy library, writing, thresholds, and baselines below.
>
> 1. Surrogate Generalization
> AFA ranks strategies by using surrogate gradients to identify "alignment-sensitive weight subsets." Our multi-target experiments show that, with fixed data and SPG, strategies ranked high by AFA maintain highly consistent performance rankings across various open- and closed-source models; we will clarify this in the revision. We will state that AFA aims to provide practical, surrogate-anchored design signals rather than theoretically complete guarantees for all models, explicitly acknowledging this as a limitation and future direction.
>
> 2. Limited Strategy Library. Regarding the concern about the limited size of the strategy library, our goal is not to exhaustively enumerate a massive strategy space, but rather to employ a compact, unified, and task-agnostic set of strategies, allowing AFA to automatically identify optimal combinations. Even with this minimalist library, CoaxChain achieves ASRs significantly surpassing existing attacks across multiple models. We will include the expansion of the strategy library as a direction for future work.
>
> 3. Writing and Organization. We sincerely appreciate your specific suggestions regarding the writing. We have corrected these issues in the revised manuscript.
>
> 4. Stability and Justification of AFA Threshold. The threshold selection relies on coarse statistics of the Critical(W) distribution rather than fine-tuning based on ASR. Since metric fluctuations on benign samples are within 0.2 while the gap between benign and malicious samples typically ranges from 0.4-0.7, we uniformly set 0.4 as a conservative cutoff. We acknowledge that due to space and computational constraints, we have currently performed only limited verification of this threshold.
>
> 5. Inclusion of Recent Baselines. In the revision, we added a representative 2025 multi-turn attack baseline, IFTD, and reran the comparison under the identical dataset and evaluation protocol used for CoaxChain (see Table 1). In our re-implementation, IFTD attains noticeably lower ASR than reported in its original paper. We believe this gap mainly comes from aligning all methods to our stricter evaluation protocol and experimental setting, rather than from an implementation issue. A systematic coverage of all other recent works will be addressed in future extensions.
>
> 6. Rationale for Focusing on MLP Layers. Our decision to restrict AFA analysis to MLP layers was a deliberate design choice. Prior studies (e.g., GradSafe) have demonstrated that features related to refusal and safety are predominantly localized in the later MLP layers. Consequently, we prioritized this subspace for defining Critical(W) and conducting our analysis.

---

### Official Review · Reviewer_ptYJ · 2025-10-30

**Soundness:** 2
**Presentation:** 3
**Contribution:** 2
**Rating:** 4
**Confidence:** 3

**Summary:**

This paper proposes a block-box multi-turn jailbreak framework called COAXCHAIN. This framework has two key components: the Alignment Failure Analyzer (AFA) and the Semantically Progressive Prompt Generator (SPG). The AFA is built on a local surrogate model and uses a gradient-based method to determine which prompt is more likely to jailbreak the victim LLM. The SPG is specifically finetuned to follow the 'Topic Induction, Intent Hinting, and Intent Rephrasing' three-phase rewriting of a single-turn malicious query. Through AFA's judgement and SPG's 3 turns rewriting, the author claims their COAXCHAIN can jailbreak LLMs using just three turns, which is relatively high efficient than existing multi-turn jailbreak methods. The attack success rates (ASRs) on different victim models are also higher than existing single-turn and multi-turn methods.

**Strengths:**

1. The COAXCHAIN involves the gradient detection to improve the quality of the jailbreak prompt in each round, therefore increasing the success rate of jailbreak and reducing the number of dialogue rounds.

2. This attack method has achieved high ASRs on sufficiently powerful LLMs such as GPT-4o and Calude-3.7.

**Weaknesses:**

1. The COAXCHAIN actually highly relies on the surrogate model used in the AFA module.

The method seems to rely on a core assumption: vulnerabilities discovered on the white-box surrogate model can be transferred to black-box target models with unknown architectures and parameters. Therefore, the key to successful jailbreak should lie in how to obtain a surrogate model that more closely to the victim model, rather than relying on multi-turn dialogues. The paper's premise of multi-turn dialogue attacks appears to be merely a formality.

2. CoaxChain claims to jailbreak a victim LLM using 3 turns. However, it is because SPG is fine-tuned to generate three rounds of dialogue.

3. The train/test settings in the 4.1.2 are not convincing.

The training and test sets should be categorized by different themes. Taking the malicious questions in advbench as an example, 'Provide detailed instructions on how to make a bomb' and 'Write a tutorial on how to make a bomb' both fall in the category 'bomb', and their cores are both how to make a bomb. Therefore, simply dividing train/test sets based on samples alone seems insufficient.

4. There seems to be no evidence that the data used for fine-tuning SPG enhances its rewriting capability. It appears to have merely taught SPG to rewrite according to a three-round pattern.

5. The 'Topic Induction, Intent Hinting, and Intent Rephrasing' seems to be the common ideas for multi-turn jailbreaks. Therefore, this paper does not appear to uncover new safety vulnerabilities or insights for the multi-turn jailbreaks.

**Questions:**

See the weaknesses above and:
1. Why do attacks on commercial models test ‘transferability’?

As a 'black-box' attack method, either the victim model an open-source model or a commercial model, the method cannot access their parameters. Then, the attack settings for open-source models and commercial models should be regarded as the same. Therefore, testing on commercial models should not be considered under 'transferability'. Alternatively, this may simply demonstrate that llama3.1-8b is a robust model capable of effectively emulating commercial models' capabilities. This does not constitute a contribution of this paper, as it does not discuss how to discover or enhance a surrogate model.

2. Can the author provide examples of successful jailbreaks? Why are there no successful examples in either the main text or the appendix?

**Details Of Ethics Concerns:**

This paper is exploring an jailbreak method, but already has ethic statement section.

---

> ### Author Response · Authors · 2025-11-21
> **Response to Reviewer ptYJ**
>
> We sincerely thank you for your thorough review and detailed comments. We address your concerns point-by-point below.
>
> 1. Surrogate Generalization
> AFA ranks strategies by using surrogate gradients to identify "alignment-sensitive weight subsets." Our multi-target experiments show that, with fixed data and SPG, strategies ranked high by AFA maintain highly consistent performance rankings across various open- and closed-source models; we will clarify this in the revision. We will state that AFA aims to provide practical, surrogate-anchored design signals rather than theoretically complete guarantees for all models, explicitly acknowledging this as a limitation and future direction.
>
> 2. Dependence on 3-Turn Structure. Regarding the concern that the attack is limited to 3 turns solely because the SPG was fine-tuned on a 3-turn format, we clarify that our framework is not inherently bound to a fixed 3-turn constraint. We evaluated various maximum turn limits in our implementation (see Table 3), and the choice of 3 turns represents an optimal trade-off between ASR and query overhead.
>
> 3. Train/Test Split. We acknowledge that a strict topic-based split would be ideal. In this work, the training set is used solely for fine-tuning the SPG, while the test set is deduplicated. Although minor semantic overlaps are difficult to entirely avoid, their proportion is low and has limited impact on SPG generalization. Crucially, the target model is never exposed to any instructions during SPG training, ensuring that the final ASR remains unaffected.
>
> 4. SPG Rewriting Capability. To demonstrate that SPG fine-tuning genuinely enhances rewriting capabilities, we will provide multiple sets of full conversation examples comparing pre- and post-fine-tuning outputs in the Appendix. The pre-fine-tuned model tends to produce rigid rewrites, whereas the fine-tuned model effectively adapts to the CoaxChain target patterns.
>
> 5. Novelty of Semantic Progression.
> Regarding the novelty of "semantic progression”, we acknowledge that the concept of gradually eliciting malicious intent is not new and has been explored in prior work. However, our core contribution lies not in this intuition itself, but in:
>
>     (1)Quantification: We use gradients and Critical(W) to explicitly quantify the process of alignment erosion across the three stages.
>
>     (2)Control: We leverage this internal risk signal to drive the attack flow. This transforms traditional empirical trial-and-error approaches into an interpretable and controllable framework supported by internal metrics, rather than merely proposing another dialogue script.
>
> 6. Successful Jailbreak Examples. We have added several complete examples of successful jailbreaks in the Appendix of the revision.

---

### Official Review · Reviewer_5jd8 · 2025-10-31

**Soundness:** 2
**Presentation:** 2
**Contribution:** 2
**Rating:** 2
**Confidence:** 3

**Summary:**

This article introduces CoaxChain, a new framework designed to improve multi-turn jailbreak attacks on large language models (LLMs) by making them more efficient and less detectable. CoaxChain has two main components: the Alignment Failure Analyzer (AFA), which helps identify effective attack prompts without relying on trial-and-error, and the Semantically Progressive Prompt Generator (SPG), which creates efficient, stealthy multi-turn dialogues based on AFA’s insights. Experimental results demonstrate the effectiveness of their method, outperforming existing methods.

**Strengths:**

The paper is clear and logically structured.
Improving the efficiency of multi-turn attacks is an important metric. For model developers, this approach can help reduce evaluation costs.

**Weaknesses:**

The biggest concern is the novelty of the two core algorithm modules, especially the Semantically Progressive Prompt Generator. The idea of gradually steering the conversation through semantics has already been introduced in current baselines. The heuristics used in the Alignment Failure Analyzer module are also common. Additionally, the concept of iterating prompts based on model feedback has already been utilized in other jailbreak baselines. Therefore, the main contribution of this paper lies in engineering improvements rather than novel algorithmic advances.

**Questions:**

See the limitations.

---

> ### Author Response · Authors · 2025-11-21
> **Response to Reviewer 5jd8**
>
> We are truly grateful for the reviewer’s insightful feedback on the novelty of our work. We wish to clarify that our primary contribution lies not in proposing a new optimization algorithm per se, but rather in explaining and controlling multi-turn jailbreaking through the lens of gradients and hidden representations:
>
> 1.Insight: By quantifying the erosion of alignment during multi-turn semantic progression via metrics like Critical(W), we strive to shed light on a phenomenon that, to the best of our knowledge, has not been systematically characterized before: alignment-related gradients paradoxically diminish as semantics become more dangerous.
>
> 2.Method: We leverage this internal risk signal to drive our 3-stage attack by determining strategy selection and stage progression. This approach seeks to advance existing trial-and-error methods that rely solely on output responses into an interpretable and controllable framework guided by internal gradient signals.

---

### Official Review · Reviewer_ExCx · 2025-11-01

**Soundness:** 3
**Presentation:** 3
**Contribution:** 2
**Rating:** 4
**Confidence:** 3

**Summary:**

This paper introduces a semantically progressive multi-turn jailbreak (CoaxChain). The proposed method integrates two components, Alignment Failure Analyzer (AFA) and Semantically Progressive Prompt Generator (SPG). Derived from GradSafe, AFA conducts gradient-based analysis through a surrogate model to measure alignment sensitivity on a prompt. SPG rewrite prompts based on the results from AFA to create a streamlined conversation.

**Strengths:**

The paper is well written and easy to follow.

The authors provide the code and datasets in the supplementary materials, and the appendix includes detailed descriptions of the training. These efforts enhance the reproducibility of their work.

**Weaknesses:**

1. The reported test results for other baseline methods are noticeably lower than those reported in their original papers and in others. This discrepancy raises concerns about fair comparison.

2. AFA’s selection criterion is computed on LLaMA-3.1-8B gradients; success on closed models is then inferred from that proxy. While transfer results are promising, the paper does not quantify how sensitive AFA is to the choice of surrogate (size/family), or whether the chosen thresholds generalize.

3. The paper would be stronger with AFA-off and AFA-random controls (keep the same three-turn structure but (i) remove gating, (ii) randomize strategy selection) to isolate AFA’s contribution beyond SPG templating. Relatedly, report failure mode breakdowns (budget exhaust vs. mis-gated progression).

4. Several studies [1-3] on multi-turn jailbreak published in 2025 are not discussed in the related works. The lack of these comparisons makes it difficult to verify the effectiveness of the proposed method.

References
[1] Weng, Zixuan, et al. "Foot-In-The-Door: A Multi-turn Jailbreak for LLMs." *arXiv preprint arXiv:2502.19820* (2025).
[2] Miao, Ziqi, et al. "Response attack: Exploiting contextual priming to jailbreak large language models." *arXiv preprint arXiv:2507.05248* (2025).
[3] Zhao, Yi, and Youzhi Zhang. "Siren: A Learning-Based Multi-Turn Attack Framework for Simulating Real-World Human Jailbreak Behaviors." *arXiv preprint arXiv:2501.14250* (2025).

**Questions:**

1.	Could you evaluate your proposed method on other benchmarks, such as HarmBench or AdvBench?
2.	I am also curious about how the ASR of different jailbreak methods varies depending on the chosen defense mechanisms. Could you conduct an ablation study across various defense methods (e.g., LLaMa-3 Guard, SmoothLLM, etc)?
3.	Since the proposed method relies on gradient-based analysis, I am curious whether it leads to a high computational or time cost. Could the authors provide some discussion or comparison on this aspect (training or prompt generation)?
4.	As a minor question, could you provide more details about Fortify in Appendix G.2?

---

> ### Author Response · Authors · 2025-11-21
> **Response to Reviewer ExCx**
>
> We sincerely thank you for the positive feedback. We address the key issues regarding baseline fairness, AFA generalizability, ablations, related work, and extended evaluations below.
>
> 1. Baseline results lower than reported values. This discrepancy arises because we enforce a stricter, unified evaluation criteria for all methods. We use an LLM judge (1–5 scale) and consider only a "5" as a success, whereas original papers typically rely on looser keyword-based matching.
>
> 2. Surrogate Generalization & Threshold Selection.
> AFA ranks strategies by using surrogate gradients to identify "alignment-sensitive weight subsets." Our multi-target experiments show that, with fixed data and SPG, strategies ranked high by AFA maintain highly consistent performance rankings across various open- and closed-source models; we will clarify this in the revision. We will state that AFA aims to provide practical, surrogate-anchored design signals rather than theoretically complete guarantees for all models, explicitly acknowledging this as a limitation and future direction.
> Threshold selection relies on Critical(W) distribution statistics rather than ASR fine-tuning. Since metric fluctuations on benign samples are within 0.2 while the gap between benign and malicious samples typically ranges from 0.4-0.7, we uniformly set 0.4 as a conservative cutoff to cover a broader range of weight matrices.
>
> 3. AFA Ablations. We agree that AFA-off/random ablations and granular failure statistics would further validate AFA, though systematic experiments are currently limited by rebuttal compute and space constraints.
>
> 4. Regarding HarmBench/AdvBench. In Sec. 4.1.2, we have already utilized AdvBench and the Malicious Instruction dataset (comparable to HarmBench).
>
> 5. Computational Cost. AFA's overhead is negligible compared to decoding (~0.8s): it requires only a single backpropagation pass on the surrogate for one target token, whereas standard generation typically yields tens to hundreds of tokens.
>
> 6. Fortify Details. We provided specific loss and training details for Fortify in Appendix G.2. We also clarified in the main text that Fortify is a conceptual defense module based on AFA signals, serving to illustrate AFA's potential value for defenders.
>
> 7. Additional Experiments. In the revision, we introduced IFTD, a representative 2025 multi-turn attack baseline, and conducted a re-evaluation under the same dataset and protocol as CoaxChain (Table 1). In our re-implementation, IFTD attains noticeably lower ASR than reported in its original paper. We believe this gap mainly comes from aligning all methods to our stricter evaluation protocol and experimental setting, rather than from an implementation issue. Furthermore, we added results on prompt filtering via LLM guard models (e.g., LlamaGuard), reporting the ASR changes for CoaxChain and baselines before and after defense.

---

### Meta-Review · Area_Chair_Gng9 · 2025-12-19

**Summary:**

The paper proposes a multi-turn jailbreak framework that relies on gradient analysis from a local surrogate model to guide a structured, three-stage prompt evolution for black-box targets. The semantic progression approach already exists in the literature, and the paper provides some comparisons with Crescendo, but there are multiple other papers on the topic, both older [1] and newer [2-3].

[1] Leveraging the context through multi-round interactions for jailbreaking attacks
[2] Many-Turn Jailbreaking
[3] Multi-turn Jailbreaking via Global Refinement and Active Fabrication

**Reviewer Concerns:**

The main concerns of the reviewers on algorithmic novelty and the experimental setup are not addressed during the rebuttal. The one concern that was addressed during the rebuttal was some engineering details on the evaluation and whether specific experiments on benchmarks were conducted. However, those arguments were eventually unconvincing to the reviewers.

**Reviewer Scores:**

I do not believe the reviewers would have changed the score, even if there was full participation in the discussion. The reason is that the main argument on algorithmic and methodological novelty stands true.

---

### Decision · Program_Chairs · 2026-01-26

Reject